# Comparing 3D, 2.5D, and 2D Approaches to Brain Image Auto-Segmentation

**DOI:** 10.3390/bioengineering10020181

**Published:** 2023-02-01

**Authors:** Arman Avesta, Sajid Hossain, MingDe Lin, Mariam Aboian, Harlan M. Krumholz, Sanjay Aneja

**Affiliations:** 1Department of Radiology and Biomedical Imaging, Yale School of Medicine, New Haven, CT 06510, USA; 2Department of Therapeutic Radiology, Yale School of Medicine, New Haven, CT 06510, USA; 3Center for Outcomes Research and Evaluation, Yale School of Medicine, New Haven, CT 06510, USA; 4Visage Imaging, Inc., San Diego, CA 92130, USA; 5Division of Cardiovascular Medicine, Yale School of Medicine, New Haven, CT 06510, USA; 6Department of Biomedical Engineering, Yale University, New Haven, CT 06510, USA

**Keywords:** auto-segmentation, deep learning, neuroimaging, magnetic resonance imaging

## Abstract

Deep-learning methods for auto-segmenting brain images either segment one slice of the image (2D), five consecutive slices of the image (2.5D), or an entire volume of the image (3D). Whether one approach is superior for auto-segmenting brain images is not known. We compared these three approaches (3D, 2.5D, and 2D) across three auto-segmentation models (capsule networks, UNets, and nnUNets) to segment brain structures. We used 3430 brain MRIs, acquired in a multi-institutional study, to train and test our models. We used the following performance metrics: segmentation accuracy, performance with limited training data, required computational memory, and computational speed during training and deployment. The 3D, 2.5D, and 2D approaches respectively gave the highest to lowest Dice scores across all models. 3D models maintained higher Dice scores when the training set size was decreased from 3199 MRIs down to 60 MRIs. 3D models converged 20% to 40% faster during training and were 30% to 50% faster during deployment. However, 3D models require 20 times more computational memory compared to 2.5D or 2D models. This study showed that 3D models are more accurate, maintain better performance with limited training data, and are faster to train and deploy. However, 3D models require more computational memory compared to 2.5D or 2D models.

## 1. Introduction

Segmentation of brain magnetic resonance images (MRIs) has widespread applications in the management of neurological disorders [1,2,3]. In patients with neurodegenerative disorders, segmenting brain structures such as the hippocampus provides quantitative information about the amount of brain atrophy [4]. In patients undergoing radiotherapy, segmentation is used to demarcate important brain structures that should be avoided to limit potential radiation toxicity [5]. Pre-operative or intra-operative brain MRIs are often used to identify important brain structures that should be avoided during neurosurgery [6,7]. Manual segmentation of brain structures on these MR images is a time-consuming task that is prone to intra- and inter-observer variability [8]. As a result, deep learning auto-segmentation methods have been increasingly used to efficiently segment important anatomical structures on brain MRIs [9].

Compared to two-dimensional (2D) auto-segmentation tasks, the three-dimensional (3D) nature of brain MRIs makes auto-segmentation considerably more challenging. There have been three proposed approaches to handling auto-segmentation of 3D images: (1) analyze and segment a two-dimensional slice of the image at a time (2D), [10] (2) analyze five consecutive two-dimensional slices at a time to generate a segmentation of the middle slice (2.5D), [11] and (3) analyze and segment the image volume in three-dimensional space (3D) [10]. Although each approach has shown some promise in medical image segmentation, a comprehensive comparison and benchmarking of these approaches for auto-segmentation of brain MRIs is lacking. Prior studies on comparing these auto-segmentation approaches have often not evaluated their efficacy in segmenting brain MRIs, or have limited their comparison narrowly to one deep learning architecture [10,12,13,14]. Additionally, previous studies have focused primarily on segmentation accuracy and failed to evaluate more practical metrics such as computational efficiency or accuracy in data-limited settings. As a result, it is difficult for clinicians and researchers to easily choose the appropriate auto-segmentation method for a desired clinical task. There is a need to compare and benchmark these three approaches for brain MRI auto-segmentation across different models and using comprehensive performance metrics.

In this study, we comprehensively compared 3D, 2.5D, and 2D approaches to brain MRI auto-segmentation across three different deep learning architectures and used metrics of accuracy and computational efficiency. We used a multi-institutional cohort of 3430 brain MRIs to train and test our models, and evaluated the efficacy of each approach across three clinically-relevant anatomical structures of the brain.

## 2. Methods

### 2.1. Dataset

This study used a dataset of 3430 T1-weighted brain MR images belonging to 841 patients from 19 institutions enrolled in the Alzheimer’s Disease Neuroimaging Initiative (ADNI) study [15]. The inclusion and exclusion criteria of ADNI have been previously described [16]. On average, each patient underwent four MRI acquisitions. Each patient underwent MR imaging using a single scanner at each site. However, the diversity of scanners in all study sites included nine different types of MR scanners. Appendix A describes the details of MRI acquisition parameters. We downloaded the anonymized MRIs of these patients from Image and Data Archive, which is a data-sharing platform [15]. The patients were randomly split into training (3199 MRIs, 93% of data), validation (117 MRIs, 3.5% of data), and test (114 MRIs, 3.5% of data) sets at the patient level. Therefore, all images belonging to a patient were assigned to either the training, validation, or test set. Table 1 summarizes patient demographics. For external validation, we additionally trained and tested a subset of our models on a dataset that contains 400 images of right and left hippocampi. The details of these experiments are provided in Appendix A.

### 2.2. Anatomic Segmentations

We trained our models to segment three representative structures of the brain: the third ventricle, thalamus, and hippocampus. These structures represent varying degrees of segmentation difficulty: the third ventricle is an easy structure to segment because it is filled with cerebrospinal fluid (CSF) with a distinct image contrast compared to surrounding structures; the thalamus is a medium-difficulty structure because it is bounded by CSF on one side and is bounded by white matter on the other side, and the hippocampus is a difficult structure because it has a complex shape and is neighbored by multiple brain structures with different image contrasts. Preliminary ground-truth segmentations were initially generated by FreeSurfer [4,17,18], and were manually corrected by a board-eligible radiologist (AA).

### 2.3. Image Pre-Processing

MRI preprocessing included corrections for B1-field variations as well as intensity inhomogeneities [19,20]. The 3D brain image was cropped around the brain after removing the skull, face, and neck tissues [21]. The input to the 3D capsule networks and 3D UNets were image patches sized 64 × 64 × 64 voxels. The inputs to the 2.5D capsule networks and 2.5D UNets were five consecutive slices of the image. The inputs to the 2D capsule networks and 2D UNets were one slice of the image. The inputs to the 3D and 2D nnUNet models were respectively 3D and 2D patches of the images with self-configured patch sizes that were automatically set by the nnUNet paradigm [22]. Appendix A describes the details of pre-processing.

### 2.4. Auto-Segmentation Models

We compared the 3D, 2.5D, and 2D approaches (Figure 1) across three segmentation models: capsule networks (CapsNets) [23], UNets [24], and nnUNets [22]. These models are considered the highest-performing auto-segmentation models in the biomedical domain [9,22,23,25,26,27,28,29]. The 3D models process a 3D patch of the image as input, all feature maps and parameter tensors in all layers are 3D, and the model output is the segmented 3D patch of the image. Conversely, 2D models process a 2D slice of the image as input, all feature maps and parameter tensors in all layers are 2D, and the model output is the segmented 2D slice of the image. The 2.5D models process five consecutive slices of the image as input channels. The remaining parts of the 2.5D model, including the feature maps and parameter tensors, are 2D, and the model output is the segmented 2D middle slice among the five slices. We did not develop 2.5D nnUNets, because the self-configuring paradigm of nnUNets was developed for 3D and 2D inputs but not for 2.5D inputs. Notably, the aim of training and testing nnUNets (in addition to UNets) was to ensure that our choices of hyperparameters did not cause one approach (such as 3D) to perform better than other approaches. The nnUNet can self-configure the best hyperparameters for the 3D and 2D approaches but not for the 2.5D approach. As a result, we did not train or test 2.5D nnUNets. The model architectures are described in Appendix A.

### 2.5. Training

We trained the CapsNet and UNet models for 50 epochs using Dice loss and the Adam optimizer [30]. Initial learning rate was set at 0.002. We used dynamic paradigms for learning rate scheduling, with a minimal learning rate of 0.0001. The hyperparameters for our CapsNet and UNet models were chosen based on the model with the lowest Dice loss over the validation set. The hyperparameters for the nnUNet model were self-configured by the model [22]. Appendix A describes the training hyperparameters for CapsNet and UNet.

### 2.6. Performance Metrics

For each model (CapsNet, UNet, and nnUNet), we compared the performance of 3D, 2.5D, and 2D approaches using the following metrics: (1) Segmentation accuracy: we used the Dice score to quantify the segmentation accuracy of the fully trained models over the test set.31 We compared Dice scores between the three approaches for three representative anatomic structures of the brain: the third ventricle, thalamus, and hippocampus. The mean Dice scores for the auto-segmentation of these brain structures are reported together with their 95% confidence interval. To compute the 95% confidence interval for each Dice score, we used bootstrapping to sample the 114 Dice scores over the test set, with replacement, 1000 times. We then calculated the mean Dice score for each of the 1000 samples, giving us 1000 mean Dice scores. We then sorted these mean Dice scores and found the range that covered 95% of them, which is equivalent to the 95% confidence interval for each Dice score. (2) Performance when training data is limited: we trained the models using the complete training set and random subsets of the training set with 600, 240, 120, and 60 MR images. The models trained on these subsets were then evaluated over the test set. (3) Computational speed during training: we compared the time needed to train the 3D, 2.5D, and 2D models per training example per epoch until the model converged. (4) Computational speed for segmenting an MR image: we compared how quickly each of the 3D, 2.5D, and 2D models segment one brain MRI volume. (5) Computational memory: we compared how much GPU memory is required, in units of megabytes, to train and deploy each of the 3D, 2.5D, and 2D models.

### 2.7. Implementation

Image pre-processing was carried out using Python (version 3.10) and FreeSurfer (version 7). PyTorch (version 1.12) was used for model development and testing. Training and testing of the models were run on GPU-equipped servers (4 vCPUs, 16 GB RAM, 16 GB NVIDIA GPU). The code used to train and test our models is available on our lab’s GitHub page: https://github.com/Aneja-Lab-Yale/Aneja-Lab-Public-3D2D-Segmentation (accessed on 4 November 2022).

## 3. Results

The segmentation accuracy of the 3D approach across all models and all anatomic structures of the brain was higher than that of the 2.5D or 2D approaches, with Dice scores of the 3D models above 90% for all anatomic structures (Table 2). Within the 3D approach, all models (CapsNet, UNet, and nnUNet) performed similarly in segmenting each anatomic structure, with their Dice scores within 1% of each other. For instance, the Dice scores of 3D CapsNet, UNet, and nnUNet in segmenting the hippocampus were respectively 92%, 93%, and 93%. Figure 2 shows auto-segmented brain structures in one patient using the three approaches. Likewise, our experiments using the external hippocampus dataset showed that 3D nnUNets achieved higher Dice scores compared to 2D nnUNets. Appendix A details the results of our experiments with the external hippocampus dataset.

3D models maintained higher accuracy, compared to 2.5D and 2D models, when training data were limited (Figure 3). When we trained the 3D, 2.5D, and 2D CapsNets using the full training set containing 3199 MRIs, their Dice scores in segmenting the third ventricle were respectively 95%, 90%, and 90%. When we trained the same models on smaller subsets of the training set containing 600, 240, 120, and 60 MRIs, the performance of 3D, 2.5D, and 2D CapsNets gradually decreased down to 90%, 88%, and 87% for the 3D, 2.5D, and 2D CapsNets, respectively (Figure 3). Importantly, the 3D CapsNet maintained higher Dice scores (over the test set) compared to 2.5D or 2D CapsNets in all these experiments. Similarly, when we trained 3D, 2.5D, and 2D UNets using the full training set, their Dice scores in segmenting the third ventricle were respectively 96%, 91%, and 90%. Decreasing the size of the training set down to 60 MRIs resulted in Dice scores of 90%, 88%, and 87% for the 3D, 2.5D, and 2D UNets, respectively. Again, the 3D UNet maintained higher Dice scores compared to 2.5D or 2D UNets in all these experiments. Lastly, when we trained 3D and 2D nnUNets using the full training set, their Dice scores in segmenting the third ventricle were respectively 96% and 90%. Decreasing the size of the training set down to 60 MRIs resulted in Dice scores of 92% and 87% for the 3D and 2D nnUNets, respectively. Once more, the 3D nnUNet maintained higher Dice scores compared to the 2D nnUNet in all these experiments (Figure 3).

The 3D models trained faster compared to 2.5D or 2D models (Figure 4). The 3D, 2.5D, and 2D CapsNets respectively took 0.8, 1, and 1 s per training example per epoch to converge during training. The 3D, 2.5D, and 2D UNets respectively took 1.6, 2.2 and 2.9 s per training example per epoch to converge during training. The 3D and 2D nnUNets respectively took 2 and 2.9 s per training example per epoch to converge during training. Therefore, 3D models converged 20% to 40% faster compared to 2.5D or 2D models. Appendix A also compares total convergence times between the 3D, 2.5D, and 2D approaches.

Fully-trained 3D models could segment brain MRIs faster during deployment compared to 2.5D or 2D models (Figure 4). Fully-trained 3D, 2.5D, and 2D CapsNets could respectively segment a brain MRI in 0.2, 0.4, and 0.4 s. Fully-trained 3D, 2.5D, and 2D UNets could respectively segment a brain MRI in 0.2, 0.3, and 0.3 s. Lastly, fully-grained 3D and 2D nnUNets could respectively segment a brain MRI in 0.3 and 0.5 s. Therefore, fully-trained 3D models segmented a brain MRI 30% to 50% faster compared to fully-trained 2.5D or 2D models.

The 3D models needed more computational memory to train and deploy as compared to the 2.5D or 2D models (Figure 5). The 3D, 2.5D, and 2D CapsNets respectively required 317, 19, and 19 megabytes of GPU memory during training. The 3D, 2.5D, and 2D UNets respectively required 3150, 180, and 180 megabytes of GPU memory. The 3D and 2D nnUNets respectively required 3200 and 190 megabytes of GPU memory. Therefore, 3D models required about 20 times more GPU memory compared to 2.5D or 2D models. Notably, CapsNets required 10 times less GPU memory compared to UNets or nnUNets. Therefore, 3D CapsNets only required two times more GPU memory compared to 2.5D or 2D UNets or nnUNets (Figure 5).

## 4. Discussion

In this study, we compared the 3D, 2.5D, and 2D approaches of auto-segmentation across three different deep learning architectures, and found that the 3D approach is more accurate, faster to train, and faster to deploy. Moreover, the 3D auto-segmentation approach maintained better performance in the setting of limited training data. We found the major disadvantage of 3D auto-segmentation approaches to be increased computational memory requirement compared to similar 2.5D and 2D auto-segmentation approaches.

Our results extend the prior literature [10,12,13,31,32,33,34] in key ways. We provide the first comprehensive benchmarking of 3D, 2.5D, and 2D approaches in auto-segmenting of brain MRIs, measuring both accuracy and computational efficiency. We compared 3D, 2.5D, and 2D approaches across three of the most successful auto-segmentation models to date, namely capsule networks, UNets, and nnUNets [22,23,26,30,33,34,35,36]. Our findings provide a practical comparison of these three auto-segmentation approaches that can provide insight when attempting auto-segmentation in settings where computational resources are bounded or when the training data are limited.

We found that the 3D approach to auto-segmentation trains faster and deploys more quickly. Previous studies that compared the computational speed of 3D and 2D UNets have concluded conflicting results: some suggested that 2D models converge faster, [10,13,32], whereas others suggested that 3D models converge faster [22]. Notably, one training iteration of 2.5D or 2D models is faster than 3D models because 2.5D and 2D models have 20 times fewer trainable parameters compared to 3D models. However, feeding a 3D image volume into a 2.5D or 2D model requires a for loop that iterates through multiple slices of the image, which slows down 2.5D and 2D models. Additionally, 3D models can converge faster during training because they can use the contextual information in the 3D image volume to segment each structure [10]. Conversely, 2.5D models can only use the contextual information in a few slices of the image [11], and 2D models can only use the contextual information in one slice only [12]. Since the 3D approach provides more contextual information for each segmentation target, the complex shape of structures such as the hippocampus can be learned faster, and, as a result, the convergence of 3D models can become faster. Lastly, each training iteration through a 3D model can be accelerated by larger GPU memory, since the training of learnable parameters can be parallelized. However, each training iteration through a 2.5D or 2D model cannot be accelerated by larger GPU memory because iterations through the slices of the image (for loop) cannot be parallelized. We propose that our findings, that 3D models converge faster, resulted from using state-of-the-art GPUs and efficient 3D models that learn contextual information in the 3D volume of the MR image faster. Our results also show that the 3D models are faster during deployment since they can process the 3D volume of the image at once, while 2.5D or 2D models must loop through 2D image slices.

Our results do highlight one of the drawbacks of 3D auto-segmentation approaches. Specifically, we found that within each model, the 3D approach requires 20 times more computational memory compared to the 2.5D or 2D approaches. Previous studies that compared 3D and 2D UNets have found similar results [10,31]. This seems to be the only downside of the 3D approach compared to the 2.5D or 2D approaches. Notably, the 2.5D approach was initially developed to achieve segmentation accuracy similar to the 3D approach while requiring computational resources similar to the 2D approach. In brain image segmentation, however, our results show that the 2.5D approach could not achieve the segmentation accuracy of the 3D approach. This raises the question of which approach to use when computational memory is limited. Our results show that *3D CapsNets* outperformed all 2.5D and 2D models while only requiring twice more computational memory than the 2.5D or 2D UNets or nnUNets. Conversely, 3D UNets and nnUNets required 20 times more computational memory compared to 2.5D or 2D UNets and nnUNets. Therefore, 3D CapsNets may be preferred in settings where computational memory is limited.

Our results corroborate previous studies showing that deep learning is accurate in biomedical image auto-segmentation [9,22,26,27,28,29]. Prior studies have shown that capsule networks, UNets, and nnUNets are the most accurate models to auto-segment biomedical images [9,11,22,23,25,26,28,33,34,36,37,38]. Prior studies have also shown that the 3D, 2.5D, and 2D versions of these models can auto-segment medical images [9,11,22,23,28,29,34]. However, evidence was lacking about which of the 3D, 2.5D, or 2D approaches would be more accurate in auto-segmenting brain structures on MR images. Our results also provide practical benchmarking of computational efficiency between the three approaches, which is often under-reported.

Our study has several notable limitations. First, we only compared the 3D, 2.5D, and 2D approaches to the auto-segmentation of brain structures on MR images. The results of this study may not generalize to other imaging modalities or other body organs. Second, there are multiple ways to develop a 2.5D auto-segmentation model [11,39,40]. While we did not implement all of the different versions of 2.5D models, we believe that our implementation of 2.5D models (using five consecutive image slices as input channels) is the best approach to segment the neuroanatomy on brain images. Third, our results about the relative deployment speed of 3D models as compared to 2.5D or 2D models might change as computational resources change. If the GPU memory is large enough to accommodate large 3D patches of the image, 3D models can segment the 3D volume faster. However, in settings where the GPU memory is limited, the 3D model should loop through multiple smaller 3D patches of the image, eroding the faster performance of the 3D models during deployment. However, we used a 16 GB GPU to train and deploy our models, which is commonplace in modern computing units used for deep learning. Finally, we compared 3D, 2.5D, and 2D approaches across three auto-segmentation models only: CapsNets, UNets, and nnUNets. While multiple other auto-segmentation models are available, we believe that our study has compared 3D, 2.5D, and 2D approaches across the most successful deep-learning models for medical image auto-segmentation. Further studies comparing the three approaches across other auto-segmentation models can be an area of future research.

## 5. Conclusions

In this study, we compared 3D, 2.5D, and 2D approaches to brain image auto-segmentation across different models and concluded that the 3D approach is more accurate, achieves better performance in the context of limited training data, and is faster to train and deploy. Our results hold across various auto-segmentation models, including capsule networks, UNets, and nnUNets. The only downside of the 3D approach is that it requires 20 times more computational memory compared to the 2.5D or 2D approaches. Because 3D capsule networks only need twice the computational memory that 2.5D or 2D UNets and nnUNets need, we suggest using 3D capsule networks in settings where computational memory is limited.

## Figures and Tables

**Figure 1 bioengineering-10-00181-f001:**
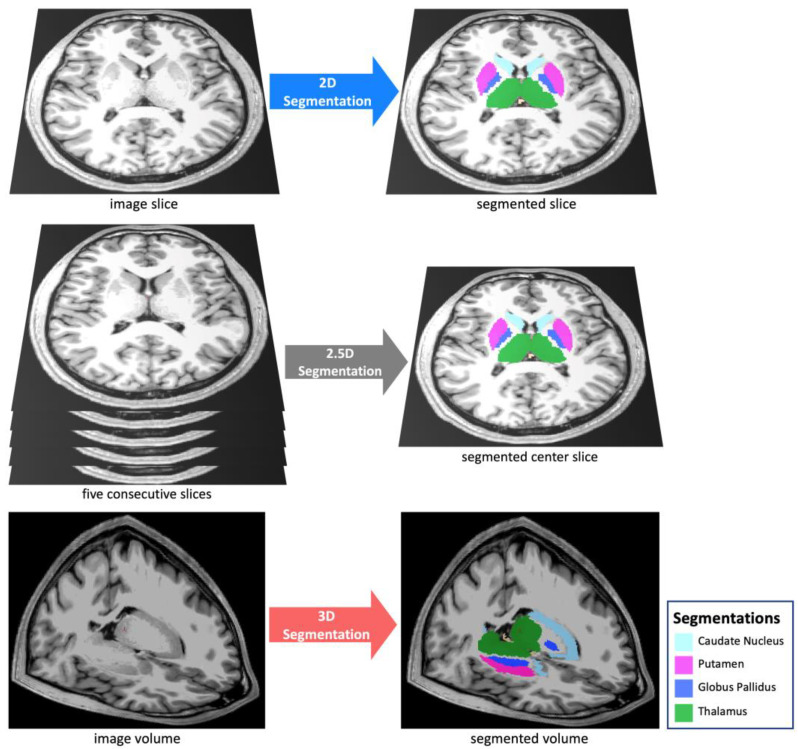
We compared three segmentation approaches: 3D, 2.5D, and 2D. The 2D approach analyzes and segments one slice of the image, the 2.5D approach analyzes five consecutive slices of the image to segment the middle slice, and the 3D approach analyzes and segments a 3D volume of the image.

**Figure 2 bioengineering-10-00181-f002:**
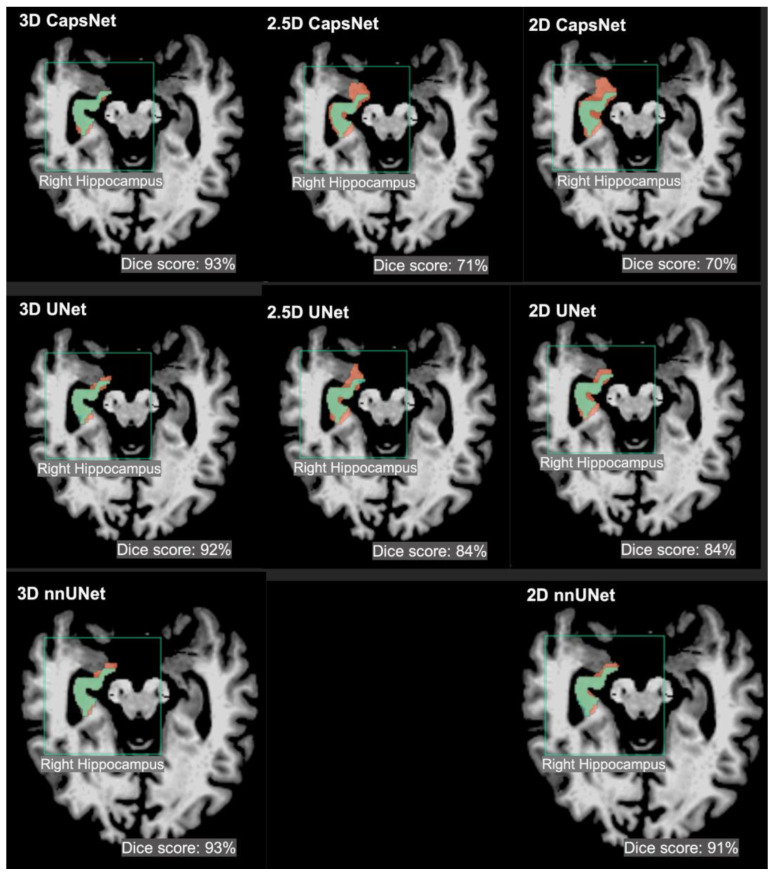
Examples of 3D, 2.5D, and 2D segmentations of the right hippocampus by CapsNet, UNet, and nnUNet. Target segmentations and model predictions are respectively shown in green and red. Dice scores are provided for the entire volume of the right hippocampus in this patient (who was randomly chosen from the test set).

**Figure 3 bioengineering-10-00181-f003:**
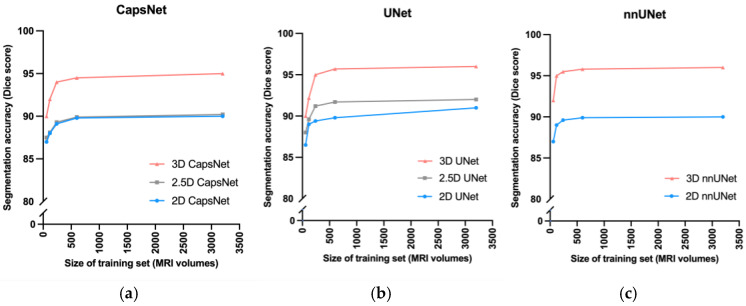
Comparing 3D, 2.5D, and 2D approaches when training data is limited. As we decreased the size of the training set from 3000 MRIs down to 60 MRIs, the CapsNet (**a**), UNet (**b**), and nnUNet (**c**) models maintained higher segmentation accuracy (measured by Dice scores).

**Figure 4 bioengineering-10-00181-f004:**
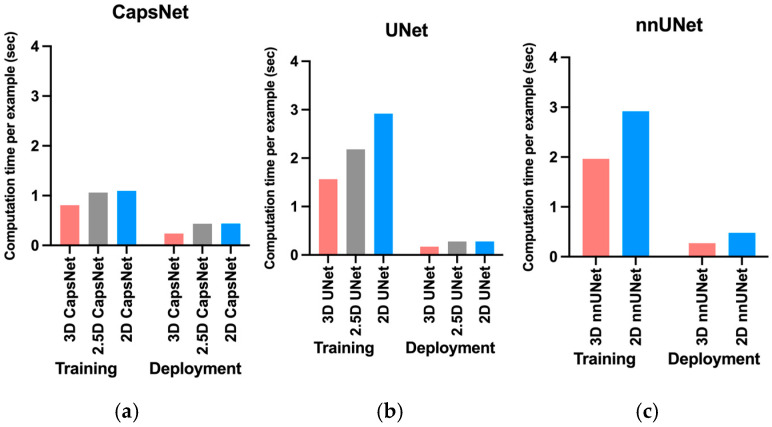
Comparing computational time required by 3D, 2.5D, and 2D approaches to train and deploy auto-segmentation models. The training times represent how much time it would take per training example per epoch for the model to converge. The deployment times represent how much time each model would require to segment one brain MRI volume. The 3D approach trained and deployed faster across all auto-segmentation models, including CapNet (**a**), UNet (**b**), and nnUNet (**c**).

**Figure 5 bioengineering-10-00181-f005:**
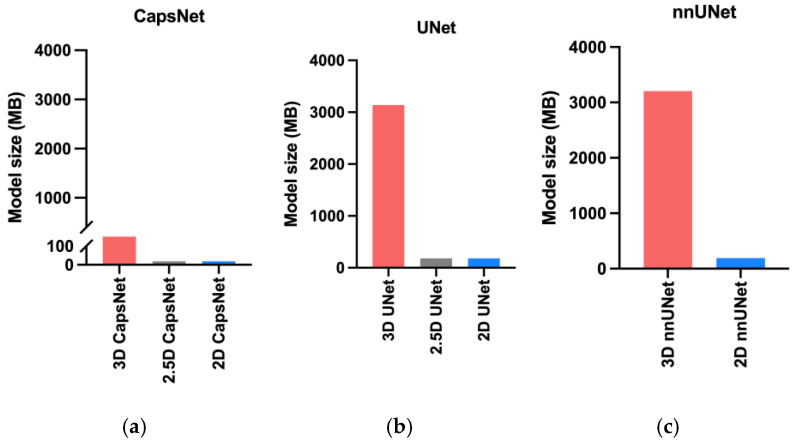
Comparing the memory required by the 3D, 2.5D, and 2D approaches. The bars represent the computational memory required to accommodate the total size of each model, including the parameters plus the cumulative size of the forward- and backward-pass feature volumes. Within each auto-segmentation model including the CapsNet (**a**), UNet (**b**), and nnUNet (**c**), the 3D approach required 20 times more computational memory compared to the 2.5D or 2D approaches.

**Table 1 bioengineering-10-00181-t001:** Study participants tabulated by the training, validation, and test sets.

Data Partitions	Number of MRIs	Number of Patients	Age (Mean ± SD)	Gender ^†^	Diagnosis ^††^
Training set	3199	841	76 ± 7	42% F, 58% M	29% CN, 54% MCI, 17% AD
Validation set	117	30	75 ± 6	30% F, 70% M	21% CN, 59% MCI, 20% AD
Test set	114	30	77 ± 7	33% F, 67% M	27% CN, 47% MCI, 26% AD

^†^ F: female; M: male. ^††^ CN: cognitively normal; MCI: mild cognitive impairment; AD: Alzheimer’s disease.

**Table 2 bioengineering-10-00181-t002:** Comparing the segmentation accuracy of 3D, 2.5D, and 2D approaches across three auto-segmentation models to segment brain structures. The three auto-segmentation models included CapsNet, UNet, and nnUNet. These models were used to segment three representative brain structures: third ventricle, thalamus, and hippocampus, which respectively represent easy, medium, and difficult structures to segment. The segmentation accuracy was quantified by Dice scores over the test (114 brain MRIs).

CapsNet			
Brain Structure	3D Dice (95% CI)	2.5D Dice (95% CI)	2D Dice (95% CI)
3rd ventricle	95% (94 to 96)	90% (89 to 91)	90% (88 to 92)
Thalamus	94% (93 to 95)	76% (74 to 78)	75% (72 to 78)
Hippocampus	92% (91 to 93)	73% (71 to 75)	71% (68 to 74)
**UNet**			
**Brain Structure**	**3D Dice (95% CI)**	**2.5D Dice (95% CI)**	**2D Dice (95% CI)**
3rd ventricle	96% (95 to 97)	92% (91 to 93)	91% (89 to 91)
Thalamus	95% (94 to 96)	92% (91 to 93)	90% (88 to 92)
Hippocampus	93% (92 to 94)	86% (84 to 88)	88% (86 to 90)
**nnUNet**	**nnUNet**	**nnUNet**	**nnUNet**
**Brain Structure**	**Brain Structure**	**Brain Structure**	**Brain Structure**
3rd ventricle	3rd ventricle	3rd ventricle	3rd ventricle
Thalamus	Thalamus	Thalamus	Thalamus
Hippocampus	Hippocampus	Hippocampus	Hippocampus

## Data Availability

The data used in this study were obtained from the Alzheimer’s Disease Neuroimaging Initiative (ADNI) database (adni.loni.usc.edu). We obtained T1-weighted MRIs of 3430 patients in the Alzheimer’s Disease Neuroimaging Initiative study from this data-sharing platform. The investigators within the ADNI contributed to the design and implementation of ADNI but did not participate in the analysis or writing of this article.

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
