# Peer review of "Comparing 3D, 2.5D, and 2D Approaches to Brain Image Auto-Segmentation"

_bioengineering, 2023, doi:10.3390/bioengineering10020181_

Round 1
Reviewer 1 Report
In the paper 3 ways of segmentation considering 3 types of input data is analysed. A dataset was divided and a learning process was carried out using 3 network architectures.
I don't notice in the body of the paper conducted cross validation of the result although the authors write in the discussion (259): We provide the first comprehensive benchmarking of 3D, 2.5D, and 2D approaches in auto-segmenting brain 260 MRIs, measuring both accuracy and computational efficiency.
The authors acknowledge elsewhere that (292) Previous studies that compared 3D and 2D UNets have found similar results.10,32
The authors write about the different levels of difficulty of recognizing (segmenting) the different elements of the studied structure, but this does not quite make itself apparent in the table of results (maybe additional figures would be beneficial).
In summary The results do not contain a sufficient elements of novelty in my opinion and for publication should be heavily revised, so I am in favor of rejecting the article in this form.
The authors also did not shy away from editing errors.
The inappropriate style of the caption under Figure 1.
Figure 1 fits the methodology rather than the introduction.
Author Response
We appreciate the reviewers’ thorough and thoughtful appraisal of our manuscript. Based on the reviewers’ comments, we have incorporated revisions to our manuscript and feel these changes have strengthened our submission.
Below, we provide a point-by-point response to the Reviewers’ specific comments.
- In the paper 3 ways of segmentation considering 3 types of input data is analysed. A dataset was divided and a learning process was carried out using 3 network architectures. I don't notice in the body of the paper conducted cross validation of the result although the authors write in the discussion: We provide the first comprehensive benchmarking of 3D, 2.5D, and 2D approaches in auto-segmenting brain 260 MRIs, measuring both accuracy and computational efficiency.
We appreciate the reviewer’s insightful commentary. We externally validated all our models using a test set, comprised of 114 MRIs, that was completely separate from the training and validation sets, as described in Methods under Datasets:
“The patients were randomly split into training (3,199 MRIs, 93% of data), validation (117 MRIs, 3.5% of data), and test (114 MRIs, 3.5% of data) sets at the patient level”.
- The authors acknowledge elsewhere that Previous studies that compared 3D and 2D UNets have found similar results.
There are indeed prior studies that have compared 3D and 2D approaches for image segmentation.1 However, no prior studies have directly compared the performance 3D, 2.5D, and 2D approaches to segment brain anatomical structures on MR images. Moreover, prior studies have shown conflicting results comparing the performance of 3D and 2D approaches.2,3 We have cited these prior studies within our introduction section.
Additionally, prior studies have notable limitations that make it difficult to draw conclusions regarding the optimal approach for segmenting brain structures on MR images. The limitations which we have addressed in our study include:
- Prior studies have not segmented clinically-relevant brain structures on MR images. Previous studies have either focused on modalities other than MRI, or structures that do not have clinical relevance.2–6 In our study, we have benchmarked segmentation techniques on MR images and paid particular attention to clinically-relevant structures which are segmented in radiation therapy planning (for example hippocampus).
- Prior studies have narrowly focused on segmentation accuracy and have not evaluated computational efficiency. Our study includes a robust evaluation of the computational memory required for each approach, which is frequently the bottleneck in training large biomedical segmentation algorithms (Figure 5 in the manuscript). Additionally, we have explored the speed of training across different segmentation approaches by measuring convergence time (Figure 4 in the manuscript).
- Prior studies have not compared the performance of 3D and 2D approaches when training data is limited.1–3,5–7 In this study, we provided evidence about the comparative performance of 3D and 2D approaches with varying degrees of limited training data (Figure 3 in the manuscript). This is an important scenario to evaluate because data is often limited in clinical data science problems.
- No prior study on auto-segmenting brain structures has included the 2.5D approach in their comparisons.8,9 As we have argued in the Discussion, the 2.5D approach was developed to use contextual information in additional image slices to segment each 2D slice, with the aim of increasing segmentation accuracy to the level of the 3D approach while keeping memory requirements at the level of the 2D approach. In this study, we extended the prior literature by showing that the 2.5D approach does not improve segmentation accuracy to the level of the 3D approach in the task of segmenting brain structures on MR images.
- The authors write about the different levels of difficulty of recognizing (segmenting) the different elements of the studied structure, but this does not quite make itself apparent in the table of results (maybe additional figures would be beneficial).
Thank you for your comment. We believe that the 3rd ventricle, thalamus, and hippocampus represent easy, medium, and difficult structures to segment for three reasons:
- Expert radiologists subjectively assign these levels of difficulty to these structures, as we have mentioned in our Methods section:
“the third ventricle is an easy structure to segment because it is filled with cerebrospinal fluid (CSF) with a distinct image contrast compared to surrounding structures; the thalamus is a medium-difficulty structure because it is bounded by CSF on one side and is bounded by white-matter on the other side, and the hippocampus is a difficult structure because it has a complex shape and is neighbored by multiple brain structures with different image contrasts”.
- Our objective results from multiple experiments confirm these levels of difficulty: within each model, as shown in Table 2, the Dice scores for the 3rd ventricle are consistently higher than the thalamus and hippocampus, and the Dice scores for the thalamus are consistently higher than the hippocampus in all our experiments. This is evident by looking at the columns of Table 2.
- The authors also did not shy away from editing errors.
We appreciate your comment. During the process of uploading the manuscript onto the journal website, some errors arose including reference formatting errors. These errors are now corrected.
- The inappropriate style of the caption under Figure 1.
Thank you for your feedback. We revised the caption of Figure 1 so that it only describes the contents of Figure 1. The revised caption is as followed:
“Figure 1. we compared three segmentation approaches: 3D, 2.5D, and 2D. The 2D approach analyzes and segments one slice of the image, the 2.5D approach analyzes five consecutive slices of the image to segment the middle slice, and the 3D approach analyzes and segments a 3D volume of the image”.
- Figure 1 fits the methodology rather than the introduction.
Thank you for your feedback. We revised the manuscript accordingly. Figure 1 is now referenced in the Methods under “Auto-segmentation Models”:
“We compared the 3D, 2.5D, and 2D approaches (Figure 1) across three segmentation models: capsule networks (CapsNets),10 UNets,11 and nnUNets.12”
References:
- Sun Y-C, Hsieh A-T, Fang S-T, et al. Can 3D artificial intelligence models outshine 2D ones in the detection of intracranial metastatic tumors on magnetic resonance images? J Chin Med Assoc JCMA 2021;84:956–62.
- Bhattacharjee R, Douglas L, Drukker K, et al. Comparison of 2D and 3D U-Net breast lesion segmentations on DCE-MRI. In: Medical Imaging 2021: Computer-Aided Diagnosis.Vol 11597. SPIE; 2021:81–7.
- Srikrishna M, Heckemann RA, Pereira JB, et al. Comparison of Two-Dimensional- and Three-Dimensional-Based U-Net Architectures for Brain Tissue Classification in One-Dimensional Brain CT. Front Comput Neurosci 2022;15.
- Kern D, Klauck U, Ropinski T, et al. 2D vs. 3D U-Net abdominal organ segmentation in CT data using organ bounds. In: Medical Imaging 2021: Imaging Informatics for Healthcare, Research, and Applications.Vol 11601. SPIE; 2021:192–200.
- Zettler N, Mastmeyer A. Comparison of 2D vs. 3D U-Net Organ Segmentation in abdominal 3D CT images. 2021 Jul 8. [Epub ahead of print].
- Nemoto T, Futakami N, Yagi M, et al. Efficacy evaluation of 2D, 3D U-Net semantic segmentation and atlas-based segmentation of normal lungs excluding the trachea and main bronchi. J Radiat Res (Tokyo) 2020;61:257–64.
- Tran M, Ly L, Hua B-S, et al. SS-3DCapsNet: Self-supervised 3D Capsule Networks for Medical Segmentation on Less Labeled Data. https://doi.org/10.48550/arXiv.2201.05905.
- Angermann C, Haltmeier M. Random 2.5D U-net for Fully 3D Segmentation. In: Vol 11794.; 2019:158–66.
- Li J, Liao G, Sun W, et al. A 2.5D semantic segmentation of the pancreas using attention guided dual context embedded U-Net. Neurocomputing 2022;480:14–26.
- Avesta A, Hui Y, Krumholz HM, et al. 3D Capsule Networks for Brain MRI Segmentation. medRxiv. https://doi.org/10.1101/2022.01.18.22269482.
- Yin X-X, Sun L, Fu Y, et al. U-Net-Based Medical Image Segmentation. J Healthc Eng 2022;2022:4189781.
- Isensee F, Jaeger PF, Kohl SAA, et al. nnU-Net: a self-configuring method for deep learning-based biomedical image segmentation. Nat Methods 2021;18:203–11.

Reviewer 2 Report
3D images require larger volume of data. The results from large volume of data must show more precise and acurate. So these results are reasoable and understandable. The scientific merit of this study is not clear for me.
Author Response
We appreciate the reviewers’ thorough and thoughtful appraisal of our manuscript. Below, we provide responses to the Reviewers’ specific comments.
- 3D images require larger volume of data. The results from large volume of data must show more precise and acurate. So these results are reasoable and understandable.
We appreciate the reviewer’s comments and agree with the reviewer that our results are reasonable and understandable.
- The scientific merit of this study is not clear for me.
There are indeed prior studies that have compared 3D and 2D approaches for image segmentation.1 However, no prior studies have directly compared the performance 3D, 2.5D, and 2D approaches to segment brain anatomical structures on MR images. Moreover, prior studies have shown conflicting results comparing the performance of 3D and 2D approaches.2,3 We have cited these prior studies within our introduction section.
Additionally, prior studies have notable limitations that make it difficult to draw conclusions regarding the optimal approach for segmenting brain structures on MR images. The limitations which we have addressed in our study include:
- Prior studies have not segmented clinically-relevant brain structures on MR images. Previous studies have either focused on modalities other than MRI, or structures that do not have clinical relevance.2–6 In our study, we have benchmarked segmentation techniques on MR images and paid particular attention to clinically-relevant structures which are segmented in radiation therapy planning (for example hippocampus).
- Prior studies have narrowly focused on segmentation accuracy and have not evaluated computational efficiency. Our study includes a robust evaluation of the computational memory required for each approach, which is frequently the bottleneck in training large biomedical segmentation algorithms (Figure 5 in the manuscript). Additionally, we have explored the speed of training across different segmentation approaches by measuring convergence time (Figure 4 in the manuscript).
- Prior studies have not compared the performance of 3D and 2D approaches when training data is limited.1–3,5–7 In this study, we provided evidence about the comparative performance of 3D and 2D approaches with varying degrees of limited training data (Figure 3 in the manuscript). This is an important scenario to evaluate because data is often limited in clinical data science problems.
- No prior study on auto-segmenting brain structures has included the 2.5D approach in their comparisons.8,9 As we have argued in the Discussion, the 2.5D approach was developed to use contextual information in additional image slices to segment each 2D slice, with the aim of increasing segmentation accuracy to the level of the 3D approach while keeping memory requirements at the level of the 2D approach. In this study, we extended the prior literature by showing that the 2.5D approach does not improve segmentation accuracy to the level of the 3D approach in the task of segmenting brain structures on MR images.
References:
- Sun Y-C, Hsieh A-T, Fang S-T, et al. Can 3D artificial intelligence models outshine 2D ones in the detection of intracranial metastatic tumors on magnetic resonance images? J Chin Med Assoc JCMA 2021;84:956–62.
- Bhattacharjee R, Douglas L, Drukker K, et al. Comparison of 2D and 3D U-Net breast lesion segmentations on DCE-MRI. In: Medical Imaging 2021: Computer-Aided Diagnosis.Vol 11597. SPIE; 2021:81–7.
- Srikrishna M, Heckemann RA, Pereira JB, et al. Comparison of Two-Dimensional- and Three-Dimensional-Based U-Net Architectures for Brain Tissue Classification in One-Dimensional Brain CT. Front Comput Neurosci 2022;15.
- Kern D, Klauck U, Ropinski T, et al. 2D vs. 3D U-Net abdominal organ segmentation in CT data using organ bounds. In: Medical Imaging 2021: Imaging Informatics for Healthcare, Research, and Applications.Vol 11601. SPIE; 2021:192–200.
- Zettler N, Mastmeyer A. Comparison of 2D vs. 3D U-Net Organ Segmentation in abdominal 3D CT images. 2021 Jul 8. [Epub ahead of print].
- Nemoto T, Futakami N, Yagi M, et al. Efficacy evaluation of 2D, 3D U-Net semantic segmentation and atlas-based segmentation of normal lungs excluding the trachea and main bronchi. J Radiat Res (Tokyo) 2020;61:257–64.
- Tran M, Ly L, Hua B-S, et al. SS-3DCapsNet: Self-supervised 3D Capsule Networks for Medical Segmentation on Less Labeled Data. https://doi.org/10.48550/arXiv.2201.05905.
- Angermann C, Haltmeier M. Random 2.5D U-net for Fully 3D Segmentation. In: Vol 11794.; 2019:158–66.
- Li J, Liao G, Sun W, et al. A 2.5D semantic segmentation of the pancreas using attention guided dual context embedded U-Net. Neurocomputing 2022;480:14–26.

Reviewer 3 Report
This is an interesting submission that considers important practical issues related to deep learning based medical image segmentation. It may be a good reference for others willing to find the best solution in terms of segmentation accuracy, limited availability of data, GPU memory and time of computation.
The text needs reformatting, especially in terms of reference numbers which should be printed as superscripts.
The main question addressed in this submission is how three selected deep-learning architectures would compare when applied to MRI brain volume segmentation tasks, in terms of segmentation accuracy, computational efficiency, size of training dataset and required GPU memory. Considering this, the title seems too general, as the research is restricted to the architectures implemented in [23] only, which do not represent all the possible 3D, 2.5D and D solutions.
In search of answers to the above question, the CapsNet, Unet and nnUnet were applied to the ADNI dataset [15], to segment the third ventricle, thalamus and hippocampus in the pre-processed images. The ground truth segmentations were obtained via FreeSurfer, then corrected by a radiologist.
This research is certainly interesting and its results form a reference for others. However, finding efficient methods of 3D brain (and other organs) image segmentation is an area of intensive work worldwide. Some of the recent studies, using modified capsule and Unet networks for medical image segmentation have been performed by Tran et al. (e.g. arXiv:2205.09299v2, 10.1109/ISBI52829.2022.9761627). Referring to those works would improve the value of this submission. Moreover, applying the CapsNet, Unet and nnUnet networks to the Hippocampus dataset (as Tran et al. did) would make the comparison even more valuable.
Round 2
Reviewer 2 Report
I found significant improvement of the manuscript.
